# Spatial-Mamba: Effective Visual State Space Models via Structure-aware State Fusion

**Chaodong Xiao**[1,2,*], **Minghan Li**[1,3,*], **Zhengqiang Zhang**[1,2], **Deyu Meng**[4], **Lei Zhang**[1,2,†]
[1]The Hong Kong Polytechnic University     [2]OPPO Research Institute
[3]Harvard Medical School     [4]Xi'an Jiaotong University
`chaodong.xiao@connect.polyu.hk, cslzhang@comp.polyu.edu.hk`
[*]Equal contribution     [†]Corresponding author

## Abstract

Selective state space models (SSMs), such as Mamba (Gu & Dao, 2023), highly excel at capturing long-range dependencies in 1D sequential data, while their applications to 2D vision tasks still face challenges. Current visual SSMs often convert images into 1D sequences and employ various scanning patterns to incorporate local spatial dependencies. However, these methods are limited in effectively capturing the complex image spatial structures and the increased computational cost caused by the lengthened scanning paths. To address these limitations, we propose **Spatial-Mamba**, a novel approach that establishes neighborhood connectivity directly in the state space. Instead of relying solely on sequential state transitions, we introduce a *structure-aware state fusion* equation, which leverages dilated convolutions to capture image spatial structural dependencies, significantly enhancing the flow of visual contextual information. Spatial-Mamba proceeds in three stages: initial state computation in a unidirectional scan, spatial context acquisition through structure-aware state fusion, and final state computation using the observation equation. Our theoretical analysis shows that Spatial-Mamba unifies the original Mamba and linear attention under the same matrix multiplication framework, providing a deeper understanding of our method. Experimental results demonstrate that Spatial-Mamba, even with a single scan, attains or surpasses the state-of-the-art SSM-based models in image classification, detection and segmentation. Source codes and trained models can be found at `https://github.com/EdwardChasel/Spatial-Mamba`.

## 1 Introduction

State space models (SSMs) are powerful tools for analyzing dynamic systems with hidden states, and they have long been utilized in fields like control theory, signal processing, and economics (Friston et al., 2003; Hafner et al., 2019; Gu et al., 2020). SSMs have been recently introduced into deep learning (Gu et al., 2021a), especially in natural language processing (NLP) (Gu & Dao, 2023), thanks to their use of specially parameterized matrices. A major advancement is the introduction of selective mechanisms and hardware-aware optimizations for parallel computing, as demonstrated by Mamba (Gu & Dao, 2023), which selectively retains or discards information based on the relevance of each element in a sequence, efficiently modeling long-distance dependencies with linear complexity.

The significant success of Mamba in NLP inspires researchers to investigate how SSMs can be applied to visual tasks. Unlike 1D sequences, visual data are typically characterized by 2D spatial structures. Therefore, it is crucial to maintain the spatial dependencies within images while adapting the information selection and propagation mechanisms in Mamba. Existing visual SSMs (Zhu et al., 2024; Liu et al., 2024; Yang et al., 2024; Huang et al., 2024; He et al., 2024a; Xiao et al., 2024) often use some scanning strategies to flatten 2D visual data into several 1D sequences from different directions, and then process the flattened 1D sequences using the original Mamba. These scanning strategies can be broadly categorized into three types: sweeping scan, continuous scan and local scan, as shown in Figs. 1(a)-1(c), respectively.

Vim (Zhu et al., 2024) and VMamba (Liu et al., 2024) employ sweeping bidirectional scan and four-way scan, which are illustrated in Fig. 1(a). These strategies aim to reduce spatial direction

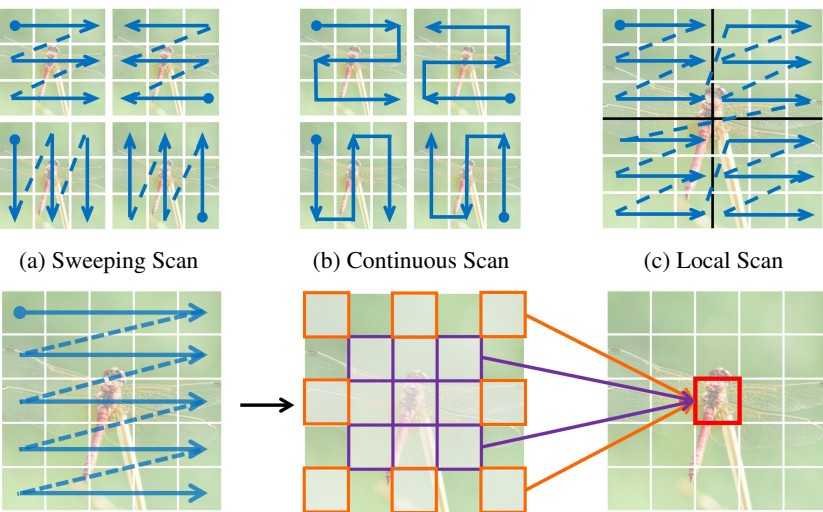

(a) Sweeping Scan  (b) Continuous Scan  (c) Local Scan

(d) Structure-aware state fusion with unidirectional scan

Figure 1: Illustration of the scanning patterns of existing visual SSMs (from sub-figures (a) to (c)) and our proposed Spatial-Mamba with structure-aware state fusion (sub-figure (d)).

sensitivity and adapt the network architecture for visual tasks. Yang et al. (2024) argued that sweeping scans neglect the importance of spatial continuity, and they introduced a continuous scanning order, as shown in Fig. 1(b), to better integrate the inductive biases from visual data. Huang et al. (2024) argued that scanning the entire image may not effectively capture local spatial relationships. Instead, they presented LocalMamba by using several local scanning modes, as shown in Fig. 1(c), which divides an image into distinct windows to capture local dependencies. Beyond the above mentioned scanning patterns, other scanning methods such as Hilbert scanning (He et al., 2024a) and dynamic tree scanning (Xiao et al., 2024) have also been proposed to adapt SSMs to visual tasks.

While the existing scanning strategies partially address the issue of aligning spatial structures with sequential SSMs, they are limited in model effectiveness and efficiency. On one hand, directional scanning inevitably alters the spatial relationships between pixels, disrupting the inherent spatial context in an image. For example, in the sweeping scan (Fig. 1(a)), the distance between a pixel and its left or right neighbor is 1, while its distance to the top or bottom neighbor equals to the image width. This distortion can hinder the models to understand the spatial relationships in the original visual data. On the other hand, the fixed scanning paths, such as the commonly used four-directional scan (Figs. 1(a) and 1(b)), are not effective enough to capture the complex and varying spatial relationships in an image, while introducing more scanning directions would also result in excessive computations. Therefore, it is imperative to explore how to design more effective and structure-aware SSMs for visual tasks.

To achieve this goal, we make an important attempt in this paper and present **Spatial-Mamba**, which is designed to capture the spatial dependencies of neighboring features in the latent state space. The processing flow of Spatial-Mamba consists of three stages. First, as shown in the left of Fig. 1(d), visual data are converted into sequential data using unidirectional sweeping scan. The state variables are computed based on the state transition equations of the original SSMs and then reshaped back into the visual format. Second, the state variables are processed through a **structure-aware state fusion** (SASF) equation, which employs dilated convolutions to re-weight and merge nearby state variables, as shown in the left of Fig. 1(d). Finally, these structure-aware state variables are fed into the observation equation to produce the final output variables. The SASF equation not only enables efficient skip connections between non-sequential elements in sequences but also enhances the model ability to capture spatial relationships, leading to more accurate representations of the underlying visual structure. Furthermore, we show that Spatial-Mamba, original Mamba and linear attention can all be represented under the same framework using structured matrices, which offers a more coherent understanding of our proposed method. We validate the superiority of Spatial-Mamba across fundamental vision tasks such as image classification, detection, and segmentation. The results demonstrate that Spatial-Mamba, even with a single scan, achieves or surpasses the performance of recent state-of-the-arts using different scanning strategies.

## 2 RELATED WORK

**State space models (SSMs).** Gu et al. (2021b) firstly introduced the linear state space layer (LSSL) into the HiPPO framework (Gu et al., 2020) to efficiently handle the long-range dependencies in long sequences. Gu et al. (2021a) then significantly improved the efficiency of SSMs by representing the parameters as diagonal plus low-rank matrix. The so-called S4 model triggers a wave of structured SSMs (Smith et al., 2022; Fu et al., 2022; Gupta et al., 2022; Gu & Dao, 2023). Smith et al. (2022) proposed S5 by introducing parallel scans to S4 layer while maintaining the computational efficiency of S4. Recently, Gu & Dao (2023) developed Mamba, which incorporates a data-dependent selection mechanism into S4 layer and simplifies the computation and architecture in a hardware-friendly way, achieving Transformer-like modeling capability with linear complexity. Building on that, Dao & Gu (2024) presented Mamba2, which reveals the connections between SSMs and attention with specific structured matrix. This framework simplifies the parameter matrix to a scalar representation, making it feasible to explore larger and more expressive state spaces without sacrificing efficiency.

**Visual SSMs.** Although traditional SSMs perform well in processing NLP sequential data and capturing temporal dependencies, they struggle in handling multi-dimensional spatial structure inherent in visual data. This limitation poses a challenge for developing effective visual SSMs. S4ND (Nguyen et al., 2022) is among the first SSM-based models for multi-dimensional data, which separates each dimension with an independent 1D SSM. Baron et al. (2023) generalized S4ND as a discrete multi-axial system and proposed the 2D-SSM spatial layer, successfully extending the 1D SSMs to 2D SSMs. More recent visual SSMs prefer to design multiple scanning orders or patterns to maintain the spatial consistency, including bidirectional (Liu et al., 2024), four-way (Zhu et al., 2024), continuous (Yang et al., 2024), zigzag (Hu et al., 2024), window-based (Huang et al., 2024), and topology-based scanning (He et al., 2024a; Xiao et al., 2024). These visual SSMs have been used in multimodal foundation models (Qiao et al., 2024; Mo & Tian, 2024), image restoration (Guo et al., 2024; Shi et al., 2024), medical image analysis (Yue & Li, 2024; Ma et al., 2024; He et al., 2024b; Liao et al., 2024) and other visual tasks (Chen et al., 2024; Li et al., 2024; Yao et al., 2024), demonstrating the potential of SSMs in visual data understanding.

## 3 PRELIMINARIES

SSMs are commonly used for analyzing sequential data and modeling continuous linear time-invariant (LTI) systems (Williams & Lawrence, 2007). An input sequence $u(t) \in \mathbb{R}$ is transformed into an output sequence $y(t) \in \mathbb{R}$ through a state variable $x(t) \in \mathbb{C}^N$. Here, $t > 0$ represents the time index, and $N$ indicates the dimension of the state variable. This dynamic system can be described by the linear state transition and observation equations (Kalman, 1960): $x'(t) = \boldsymbol{A}x(t) + \boldsymbol{B}u(t), y(t) = \boldsymbol{C}x(t) + \boldsymbol{D}u(t)$, where $\boldsymbol{A} \in \mathbb{C}^{N \times N}$ is the state transition matrix, $\boldsymbol{B}, \boldsymbol{C} \in \mathbb{C}^N$ and $\boldsymbol{D} \in \mathbb{C}^1$ control the dynamics of the system. The state transition and observation equations describe how the system evolves over time and how the state variables relate to the observed outputs.

To effectively integrate continuous-time SSMs into the deep learning framework, it is essential to discretize the continuous-time models. One commonly employed technique is the Zero-Order Hold (ZOH) discretization (Gu & Dao, 2023). The ZOH method approximates the continuous-time system by holding the input constant over each discrete time interval. Specifically, given a timescale $\boldsymbol{\Delta}$, which represents the interval between discrete time steps, and defining $\overline{\boldsymbol{A}}$ and $\overline{\boldsymbol{B}}$ as discrete parameters, the ZOH rule is applied as $\overline{\boldsymbol{A}} = e^{\boldsymbol{\Delta A}}$ and $\overline{\boldsymbol{B}} = (\boldsymbol{\Delta A})^{-1}(e^{\boldsymbol{\Delta A}} - I)\boldsymbol{\Delta B}$.

However, real-world processes often change over time and cannot be accurately described by a LTI system. As highlighted in Mamba (Gu & Dao, 2023), time-varying systems can focus more on relevant information and offer a more accurate and realistic representation of dynamic systems. In Mamba, the parameters of the SSMs are made context-aware and adaptive through selective functions. This is achieved by modifying the parameters $\boldsymbol{\Delta}, \boldsymbol{B}, \boldsymbol{C}$ as simple functions of the input sequence $u_t$, resulting in input-dependent parameters $\boldsymbol{\Delta}_t = s_\Delta(u_t), \boldsymbol{B}_t = s_B(u_t)$ and $\boldsymbol{C}_t = s_C(u_t)$. Then the input-dependent discrete parameters $\overline{\boldsymbol{A}}_t$ and $\overline{\boldsymbol{B}}_t$ can be calculated accordingly. Consequently, the discrete state transition and observation equations can be calculated as follows:

$$x_t = \overline{\boldsymbol{A}}_t x_{t-1} + \overline{\boldsymbol{B}}_t u_t, \quad y_t = \boldsymbol{C}_t x_t + \boldsymbol{D}u_t. \tag{1}$$

A simplified illustration of the above process is shown in Fig. 2(a).

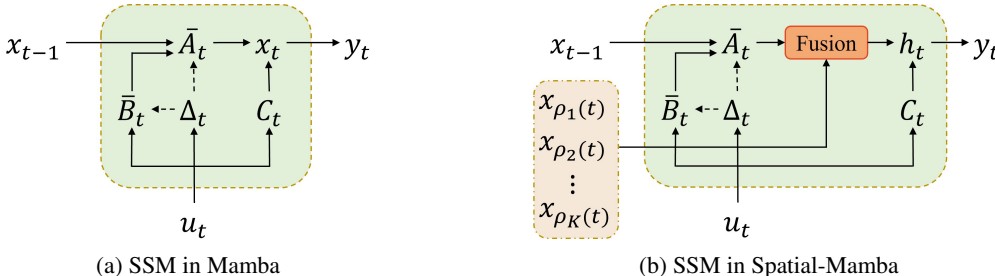

(a) SSM in Mamba     (b) SSM in Spatial-Mamba

Figure 2: Illustrations of the SSM in (a) Mamba and (b) our Spatial-Mamba, where the residual term $\boldsymbol{D}$ is omitted. In (b), 'Fusion' refers to our proposed structure-aware state fusion (SASF) equation.

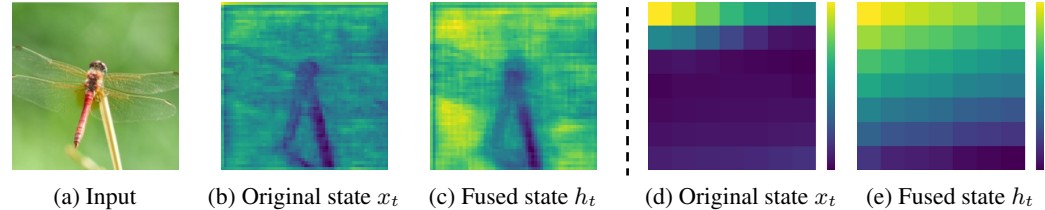

(a) Input   (b) Original state $x_t$   (c) Fused state $h_t$   (d) Original state $x_t$   (e) Fused state $h_t$

Figure 3: Visualization of state variables before and after applying the SASF equation. Sub-figures (b) and (c) show the mean of state variables across all channels in the first layer of Spatial-Mamba, while sub-figures (d) and (e) display the state variables for a specific channel in the last layer.

## 4 SPATIAL-MAMBA

### 4.1 FORMULATION OF SPATIAL-MAMBA

Spatial-Mamba is designed to capture the spatial dependencies of neighboring features in the latent state space. To achieve this goal, different from previous methods (Zhu et al., 2024; Liu et al., 2024; Huang et al., 2024) that commonly employ multiple scanning directions, we introduce a new structure-aware state fusion (SASF) equation into the original Mamba formulas (refer to Eq. (1)). The entire process of Spatial-Mamba can be described by three equations: the state transition equation, the SASF and the observation equation, which are formulated as:

$$x_t = \overline{\boldsymbol{A}}_t x_{t-1} + \overline{\boldsymbol{B}}_t u_t, \quad h_t = \sum_{k \in \Omega} \alpha_k x_{\rho_k(t)}, \quad y_t = \boldsymbol{C}_t h_t + \boldsymbol{D} u_t, \tag{2}$$

where $x_t$ is the original state variable, $h_t$ is the structure-aware state variable, $\Omega$ is the neighbor set, $\alpha_k$ is a learnable weight, and $\rho_k(t)$ is the index of the $k$-th neighbor of position $t$. Fig. 2(b) illustrates the SSM flow in the proposed Spatial-Mamba. Compared with the original Mamba in Fig. 2(a), we can see that the original state variable $x_t$ is directly influenced by its previous state $x_{t-1}$, while the structure-aware state variable $h_t$ incorporates additional neighboring state variables $x_{\rho_1(t)}, x_{\rho_2(t)}, \ldots, x_{\rho_K(t)}$ through a fusion mechanism, where $K = |\Omega|$ denotes the size of neighbor set. By considering both the global long-range and the local spatial information, the fused state variable $h_t$ gains a richer context, leading to improved adaptability and a more comprehensive understanding of the image.

The proposed Spatial-Mamba can be implemented in three steps. As shown in Fig. 1(d), the input image is first flattened into a 1D sequence $u_t$, with which the state $x_t$ is computed using the state transition equation $x_t = \overline{\boldsymbol{A}}_t x_{t-1} + \overline{\boldsymbol{B}}_t u_t$. The computed states are then reshaped back into the 2D format. To enable each state to be aware of its neighboring states in the 2D space, we introduce the SASF equation $h_t = \sum_{k \in \Omega} \alpha_k x_{\rho_k(t)}$. For a state variable $x_t$, we apply linear weighting to its neighboring states $\rho_k(t)$ in the neighborhood $\Omega$ using weights $\alpha_k$ so that we can effectively integrate local dependency information into a new state $h_t$, resulting in a more contextually informative representation. Finally, the output is generated from this enriched state $h_t$ through the observation equation $y_t = \boldsymbol{C}_t h_t + \boldsymbol{D} u_t$. This SASF approach helps the model to incorporate the local structural information in visual learning while retaining the benefits of original Mamba.

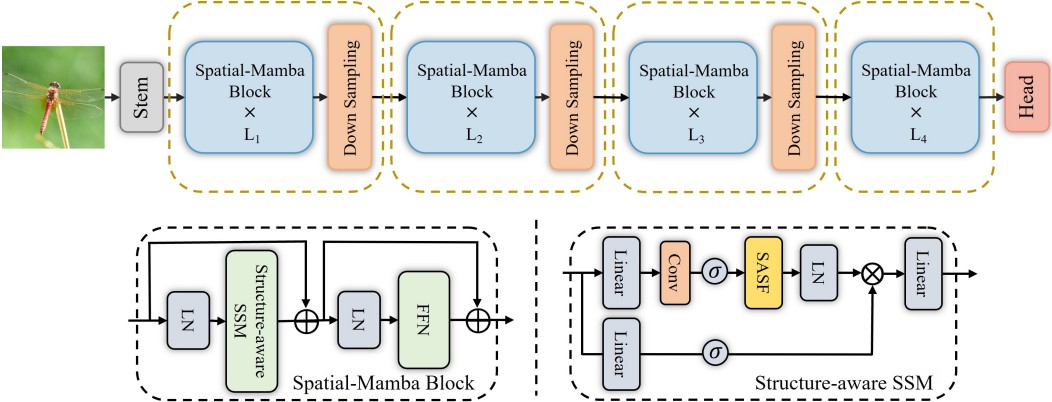

Figure 4: Overall network architecture of Spatial-Mamba.

In practice, we simply employ multi-scale dilated convolutions (Yu, 2015) to linearly weight adjacent state variables, enhancing spatial relationship characterizations and enabling skip connections. Specifically, we use three $3 \times 3$ depth-wise filters with dilation factors $d = 1, 3, 5$ to construct the neighbor set $\Omega_d = \{(i, j) | i, j \in \{-d, 0, d\}\}$. The SASF equation in Eq. (2) can be rewritten as:

$$h_t = \sum_{d=1,3,5} \sum_{i,j \in \Omega_d} k_{ij}^d \cdot x_{t+iw+j}, \tag{3}$$

where $k_{ij}^d$ represents the filter weight for dilation factor $d$ at position $(i, j)$, $x_{t+iw+j}$ denotes the neighbor of the state $x_t$ located at the position $(i, j)$, and $w$ indicates the width of the image.

To gain an intuitive understanding of SASF, we visualize the original state variables and our proposed structure-aware state variables in Fig. 3. Additional visualizations are provided in Appendix A. One can see that the original state $x_t$ (Fig. 3(b)) struggles to differentiate the foreground from background. In contrast, the structure-aware state $h_t$ (Fig. 3(c)), which has been refined through a SASF process, effectively separates these regions. Moreover, the original state variable $x_t$ in Fig. 3(d) only shows horizontal attenuation along the scanning direction (gradually darkening from the brightest value in the upper left corner), while the fused state variable $h_t$ in Fig. 3(e) demonstrates decay along the horizontal, vertical and diagonal directions. This improvement stems from its ability to leverage spatial relationships within the image, leading to a more accurate and context-aware representation.

## 4.2 NETWORK ARCHITECTURE

The overall architecture of Spatial-Mamba is depicted in Fig. 4. It consists of four successive stages, resembling the structure of Swin-Transformer (Liu et al., 2021). We introduce three variants of Spatial-Mamba model at different scales: Spatial-Mamba-T (tiny), Spatial-Mamba-S (small), and Spatial-Mamba-B (base). The detailed configurations are provided in Appendix B. Specifically, an input image $I \in \mathbb{R}^{H \times W \times 3}$ is first processed by an overlapped stem layer to generate a 2D feature map with dimension of $\frac{H}{4} \times \frac{W}{4} \times C$. This feature map is then fed into four successive stages. Each stage comprises multiple Spatial-Mamba blocks, followed by a down-sampling layer with a factor of 2 (except for the last stage), resulting in hierarchical features. Finally, a head layer is employed to process these features to produce corresponding image representations for specific downstream tasks.

The Spatial-Mamba block forms the fundamental building unit of our architecture, which consists of a Structure-aware SSM and a Feed-Forward Network (FFN) with skip connections, as illustrated in the bottom left of Fig. 4. Building upon the Mamba block design (Gu & Dao, 2023), the Structure-aware SSM, illustrated in the bottom right of Fig. 4, is implemented by substituting the original 1D causal convolution with a $3 \times 3$ depth-wise convolution and replacing the original S6 module with our proposed SASF module, achieving local neighborhood connectivity in state spaces with linear complexity. Moreover, a local perception unit (LPU) (Guo et al., 2022) is employed before the Spatial-Mamba block and FFN to extract local information inside image patches.

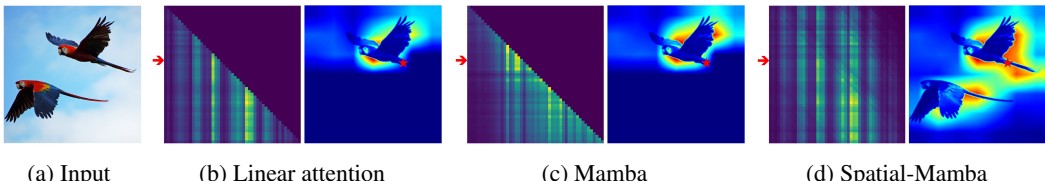

| (a) Input | (b) Linear attention | (c) Mamba | (d) Spatial-Mamba |

Figure 5: Visualizations of matrices $M$ and the corresponding activation maps for linear attention, Mamba and Spatial-Mamba. The red arrows indicate specific rows in matrices $M$, along with the corresponding image patches (marked with a red star).

## 4.3 CONNECTION WITH ORIGINAL MAMBA AND LINEAR ATTENTION

We analyze in-depth the similarities and disparities among linear attention (Katharopoulos et al., 2020), original Mamba (Gu & Dao, 2023) and our Spatial-Mamba, providing a better understanding of the working mechanism of our proposed method. Detailed derivations are provided in Appendix C.

**Linear attention** is an improved self-attention (SA) mechanism, reducing the computational complexity of SA to linear by using a kernel function $\phi$. For an input sequence $u_t$, the query $q_t$, key $k_t$, and value $v_t$ are computed by projecting $u_t$ with different weight matrices. In autoregressive models, to prevent the model from attending to future tokens, the $t$-th query is restricted by the previous keys, *i.e.*, $k_s, s \leq t$. Thus, if the kernel function $\phi$ is an identity map, the calculation of single head linear attention without normalization can be formulated as: $y_t = \sum_{s \leq t} q_t(k_s^T v_s)$. Letting $x_t = \sum_{s \leq t} k_s^T v_s$, then we have $x_t = x_{t-1} + k_t^T v_t$. The linear attention can be rewritten as $y_t = q_t x_t$. This reveals that linear attention is actually a special case of linear recursion. The hidden state variable $x_t$ is updated by the outer-product $k_s^T v_s$, and the final output $y_t$ is observed by multiplying $x_t$ with the query $q_t$. If we define $C_t = q_t$ and $\overline{B}_s = k_s^T$, the linear attention can be expressed in a form similar to that of SSM:

$$y_t = \sum_{s \leq t} C_t \overline{B}_s v_s, \tag{4}$$

where $v_s$ is a linear transformation of $u_s$, *i.e.*, $v_s = \text{Linear}(u_s)$.

**Mamba** is essentially defined in Eq. (1). By setting the initial state variable $x_0$ to zero, the state variables can be derived recursively as $x_t = \sum_{s \leq t} \overline{A}_{s:t}^{\times} \overline{B}_s u_s$. Here, $\overline{A}_{s:t}^{\times} := \Pi_{i=s+1}^{t} \overline{A}_i$ denotes the product of the state transition matrices with indices from $s+1$ to $t$ for $s < t$, and its value is 1 when $s = t$. The final output of the observation equation without $Du_t$ can be rewritten as:

$$y_t = \sum_{s \leq t} C_t \overline{A}_{s:t}^{\times} \overline{B}_s u_s. \tag{5}$$

**Spatial-Mamba.** Based on Eq. (2) and Eq. (5), the structure-aware state variables from our proposed SASF equation can be expressed as $h_t = \sum_{k \in \Omega} \sum_{s \leq \rho_k(t)} \alpha_k \overline{A}_{s:\rho_k(t)}^{\times} \overline{B}_s u_s$. By omitting $Du_t$ for simplicity of expression, the final output of Spatial-Mamba can be reformulated as follows:

$$y_t = \sum_{k \in \Omega} \sum_{s \leq \rho_k(t)} \alpha_k C_t \overline{A}_{s:\rho_k(t)}^{\times} \overline{B}_s u_s. \tag{6}$$

**Remarks.** From the above analysis, we can see that all the three paradigms — linear attention, Mamba, and Spatial-Mamba — can be modeled within a unified matrix multiplication framework, specifically $y = Mu$. The differences lie in the structure of $M$. For both linear attention and Mamba, $M$ takes the form of a **lower triangular matrix**, whereas for Spatial-Mamba, $M$ is an **adjacency matrix**. Fig. 5 provides a visualization of these matrices. In linear attention, the positions of brighter values remain consistent along the vertical direction, which indicates that the SA mechanism puts its focus on a small set of image tokens. Mamba, on the other hand, shows a decaying pattern over time, which is attributed to the influence of its state transition matrix $\overline{A}_t$. This dynamic transition allows Mamba to shift its focus among previous image tokens. Unlike linear attention and Mamba, our Spatial-Mamba considers the weighted summation of all states within a broader spatial neighborhood $\Omega$, allowing for a more comprehensive representation of spatial relationships. The activation maps on the right side of Fig. 5 further demonstrate that linear attention focuses on a limited region, while Mamba captures a broader region due to its long-range context modeling. Our Spatial-Mamba not only largely extends the range of context modeling but also enables spatial structure modeling, effectively identifying relevant regions even when they are distant from each other.

## 5 EXPERIMENTAL RESULTS

In this section, we conduct a series of experiments to compare Spatial-Mamba with leading benchmark models, including those based on Convolutional Neural Networks (CNNs) (Radosavovic et al., 2020; Liu et al., 2022; Yu & Wang, 2024), Vision Transformers (Dosovitskiy et al., 2020; Liu et al., 2021; Wang et al., 2022; Hassani et al., 2023), and recent visual SSMs (Nguyen et al., 2022; Zhu et al., 2024; Liu et al., 2024; Huang et al., 2024). Following previous works (Liu et al., 2021; 2024), we train three variants of Spatial-Mamba, namely Spatial-Mamba-T, Spatial-Mamba-S and Spatial-Mamba-B. The detailed configurations are provided in Appendix B. The performance evaluation is conducted on fundamental visual tasks, including image classification, object detection, and semantic segmentation.

### 5.1 IMAGE CLASSIFICATION ON IMAGENET-1K

**Settings.** We first evaluate the representation learning capabilities of Spatial-Mamba in image classification on ImageNet-1K (Deng et al., 2009). We adopted the experimental configurations used in previous works (Liu et al., 2021; 2024), which are detailed in Appendix B. We compare our method with state-of-the-art approaches, including RegNetY (Radosavovic et al., 2020), ConvNeXt (Liu et al., 2022), ViT (Dosovitskiy et al., 2020), DeiT (Touvron et al., 2021), Swin (Liu et al., 2021), NAT (Hassani et al., 2023), S4ND (Nguyen et al., 2022), Vim (Zhu et al., 2024), VMamba (Liu et al., 2024), and LocalVMamba (Huang et al., 2024).

Table 1: Comparison of classification performance on ImageNet-1K, where 'Throughput' is measured using an A100 GPU with an input resolution of $224 \times 224$.

| Arch. | Method | Im. size | #Param. (M) | FLOPs (G) | Throughput↑ | Top-1 acc.↑ |
|---|---|---|---|---|---|---|
| CNN | RegNetY-4G | $224^2$ | 21M | 4.0G | - | 80.0 |
| | RegNetY-8G | $224^2$ | 39M | 8.0G | - | 81.7 |
| | RegNetY-16G | $224^2$ | 84M | 16.0G | - | 82.9 |
| | ConvNeXt-T | $224^2$ | 29M | 4.5G | 1189 | 82.1 |
| | ConvNeXt-S | $224^2$ | 50M | 8.7G | 682 | 83.1 |
| | ConvNeXt-B | $224^2$ | 89M | 15.4G | 435 | 83.8 |
| Transformer | ViT-B/16 | $384^2$ | 86M | 55.4G | - | 77.9 |
| | ViT-L/16 | $384^2$ | 307M | 190.7G | - | 76.5 |
| | DeiT-S | $224^2$ | 22M | 4.6G | 1759 | 79.8 |
| | DeiT-B | $224^2$ | 86M | 17.5G | 500 | 81.8 |
| | DeiT-B | $384^2$ | 86M | 55.4G | 498 | 83.1 |
| | Swin-T | $224^2$ | 28M | 4.5G | 1244 | 81.3 |
| | Swin-S | $224^2$ | 50M | 8.7G | 718 | 83.0 |
| | Swin-B | $224^2$ | 88M | 15.4G | 458 | 83.5 |
| | NAT-T | $224^2$ | 28M | 4.3G | - | 83.2 |
| | NAT-S | $224^2$ | 51M | 7.8G | - | 83.0 |
| | NAT-B | $224^2$ | 90M | 13.7G | - | 84.3 |
| SSM | S4ND-ConvNeXt-T | $224^2$ | 30M | - | 683 | 82.2 |
| | S4ND-ViT-B | $224^2$ | 89M | - | 397 | 80.4 |
| | ViM-S | $224^2$ | 26M | - | 811 | 80.5 |
| | VMamba-T | $224^2$ | 30M | 4.9G | 1686 | 82.6 |
| | VMamba-S | $224^2$ | 50M | 8.7G | 877 | 83.6 |
| | VMamba-B | $224^2$ | 89M | 15.4G | 646 | 83.9 |
| | LocalVMamba-T | $224^2$ | 26M | 5.7G | 394 | 82.7 |
| | LocalVMamba-S | $224^2$ | 50M | 11.4G | 227 | 83.7 |
| | Spatial-Mamba-T | $224^2$ | 27M | 4.5G | 1438 | **83.5** |
| | Spatial-Mamba-S | $224^2$ | 43M | 7.1G | 988 | **84.6** |
| | Spatial-Mamba-B | $224^2$ | 96M | 15.8G | 665 | **85.3** |

Table 2: Comparison of object detection and instance segmentation performance on COCO with Mask R-CNN (He et al., 2017) detector. FLOPs are calculated with input resolution of $1280 \times 800$.

| Backbone | $AP^b \uparrow$ | $AP^b_{50} \uparrow$ | $AP^b_{75} \uparrow$ | $AP^m \uparrow$ | $AP^m_{50} \uparrow$ | $AP^m_{75} \uparrow$ | #Param. | FLOPs |
|---|---|---|---|---|---|---|---|---|
| **Mask R-CNN 1× schedule** | | | | | | | | |
| ResNet-50 | 38.2 | 58.8 | 41.4 | 34.7 | 55.7 | 37.2 | 44M | 260G |
| Swin-T | 42.7 | 65.2 | 46.8 | 39.3 | 62.2 | 42.2 | 48M | 267G |
| ConvNeXt-T | 44.2 | 66.6 | 48.3 | 40.1 | 63.3 | 42.8 | 48M | 262G |
| PVTv2-B2 | 45.3 | 66.1 | 49.6 | 41.2 | 64.2 | 44.4 | 45M | 309G |
| ViT-Adapter-S | 44.7 | 65.8 | 48.3 | 39.9 | 62.5 | 42.8 | 48M | 403G |
| MambaOut-T | 45.1 | 67.3 | 49.6 | 41.0 | 64.1 | 44.1 | 43M | 262G |
| VMamba-T | 47.3 | 69.3 | 52.0 | 42.7 | 66.4 | 45.9 | 50M | 271G |
| LocalVMamba-T | 46.7 | 68.7 | 50.8 | 42.2 | 65.7 | 45.5 | 45M | 291G |
| Spatial-Mamba-T | **47.6** | **69.6** | **52.3** | **42.9** | **66.5** | **46.2** | 46M | 261G |
| ResNet-101 | 38.2 | 58.8 | 41.4 | 34.7 | 55.7 | 37.2 | 63M | 336G |
| Swin-S | 44.8 | 68.6 | 49.4 | 40.9 | 65.3 | 44.2 | 69M | 354G |
| ConvNeXt-S | 45.4 | 67.9 | 50.0 | 41.8 | 65.2 | 45.1 | 70M | 348G |
| PVTv2-B3 | 47.0 | 68.1 | 51.7 | 42.5 | 65.2 | 45.7 | 63M | 397G |
| MambaOut-S | 47.4 | 69.1 | 52.4 | 42.7 | 66.1 | 46.2 | 65M | 354G |
| VMamba-S | 48.7 | 70.0 | 53.4 | 43.7 | 67.3 | 47.0 | 70M | 349G |
| LocalVMamba-S | 48.4 | 69.9 | 52.7 | 43.2 | 66.7 | 46.5 | 69M | 414G |
| Spatial-Mamba-S | **49.2** | **70.8** | **54.2** | **44.0** | **67.9** | **47.5** | 63M | 315G |
| Swin-B | 46.9 | - | - | 42.3 | 66.3 | 46.0 | 88M | 496G |
| ConvNeXt-B | 47.0 | 69.4 | 51.7 | 42.7 | 66.3 | 46.0 | 107M | 486G |
| PVTv2-B5 | 47.4 | 68.6 | 51.9 | 42.5 | 65.7 | 46.0 | 102M | 557G |
| ViT-Adapter-B | 47.0 | 68.2 | 51.4 | 41.8 | 65.1 | 44.9 | 102M | 557G |
| MambaOut-B | 47.4 | 69.3 | 52.2 | 43.0 | 66.4 | 46.3 | 100M | 495G |
| VMamba-B | 49.2 | 71.4 | 54.0 | 44.1 | 68.3 | 47.7 | 108M | 485G |
| Spatial-Mamba-B | **50.4** | **71.8** | **55.3** | **45.1** | **69.1** | **49.1** | 115M | 494G |
| **Mask R-CNN 3× MS schedule** | | | | | | | | |
| Swin-T | 46.0 | 68.1 | 50.3 | 41.6 | 65.1 | 44.9 | 48M | 267G |
| ConvNeXt-T | 46.2 | 67.9 | 50.8 | 41.7 | 65.0 | 44.9 | 48M | 262G |
| NAT-T | 47.7 | 69.0 | 52.6 | 42.6 | 66.1 | 45.9 | 48M | 258G |
| VMamba-T | 48.8 | 70.4 | 53.5 | 43.7 | 67.4 | 47.0 | 50M | 271G |
| LocalVMamba-T | 48.7 | 70.1 | 53.0 | 43.4 | 67.0 | 46.4 | 45M | 291G |
| Spatial-Mamba-T | **49.3** | **70.7** | **54.3** | **43.8** | **67.8** | **47.2** | 46M | 261G |
| Swin-S | 48.2 | 69.8 | 52.8 | 43.2 | 67.0 | 46.1 | 69M | 354G |
| ConvNeXt-S | 47.9 | 70.0 | 52.7 | 42.9 | 66.9 | 46.2 | 70M | 348G |
| NAT-S | 48.4 | 69.8 | 53.2 | 43.2 | 66.9 | 46.5 | 70M | 330G |
| VMamba-S | 49.9 | 70.9 | 54.7 | 44.2 | 68.2 | 47.7 | 70M | 349G |
| LocalVMamba-S | 49.9 | 70.5 | 54.4 | 44.1 | 67.8 | 47.4 | 69M | 414G |
| Spatial-Mamba-S | **50.5** | **71.5** | **55.5** | **44.6** | **68.7** | **47.8** | 63M | 315G |

**Results.** Tab. 1 presents a comprehensive comparison between Spatial-Mamba against state-of-the-art methods. Notably, Spatial-Mamba-T achieves a top-1 accuracy of 83.5%, outperforming the CNN-based ConvNext-T by 1.4% with similar amount of parameters and FLOPs. Compared to Transformer-based methods, Spatial-Mamba-T exceeds Swin-T by 2.2% and NAT-T by 0.3%. In comparison with SSM-based methods, Spatial-Mamba-T outperforms VMamba-T by 1.0% and LocalVMamba-T by 0.8%. For other variants, Spatial-Mamba also shows advantages. Specifically, Spatial-Mamba-S and Spatial-Mamba-B achieve top-1 accuracies of 84.6% and 85.3%, respectively, surpassing NAT-S and NAT-B by margins of 1.6% and 1.0%, and VMamba-S and VMamba-B by 1.0% and 1.4%. While Spatial-Mamba-T is slightly slower than VMamba-T due to architectural differences, the Small and Base variants of Spatial-Mamba are faster than their VMamba counterparts. Moreover, both of them are significantly faster than CNN and Transformer-based methods.

Table 3: Comparison of semantic segmentation on ADE20K with UPerNet (Xiao et al., 2018) segmentor. FLOPs are calculated with input resolution of $512 \times 2048$. 'SS' and 'MS' represent single-scale and multi-scale testing, respectively.

| Method | Crop size | mIoU (SS) ↑ | mIoU (MS) ↑ | #Param. | FLOPs |
|---|---|---|---|---|---|
| DeiT-S + MLN | $512^2$ | 43.1 | 43.8 | 58M | 1217G |
| Swin-T | $512^2$ | 44.4 | 45.8 | 60M | 945G |
| ConvNeXt-T | $512^2$ | 46.0 | 46.7 | 60M | 939G |
| NAT-T | $512^2$ | 47.1 | 48.4 | 58M | 934G |
| MambaOut-T | $512^2$ | 47.4 | 48.6 | 54M | 938G |
| VMamba-T | $512^2$ | 48.0 | 48.8 | 62M | 949G |
| LocalVMamba-T | $512^2$ | 47.9 | 49.1 | 57M | 970G |
| Spatial-Mamba-T | $512^2$ | **48.6** | **49.4** | 57M | 936G |
| DeiT-B + MLN | $512^2$ | 45.5 | 47.2 | 144M | 2007G |
| Swin-S | $512^2$ | 47.6 | 49.5 | 81M | 1039G |
| ConvNeXt-S | $512^2$ | 48.7 | 49.6 | 82M | 1027G |
| NAT-S | $512^2$ | 48.0 | 49.5 | 82M | 1010G |
| MambaOut-S | $512^2$ | 49.5 | 50.6 | 76M | 1032G |
| VMamba-S | $512^2$ | **50.6** | 51.2 | 82M | 1028G |
| LocalVMamba-S | $512^2$ | 50.0 | 51.0 | 81M | 1095G |
| Spatial-Mamba-S | $512^2$ | **50.6** | **51.4** | 73M | 992G |
| Swin-B | $512^2$ | 48.1 | 49.7 | 121M | 1188G |
| ConvNeXt-B | $512^2$ | 49.1 | 49.9 | 122M | 1170G |
| NAT-B | $512^2$ | 48.5 | 49.7 | 123M | 1137G |
| MambaOut-B | $512^2$ | 49.6 | 51.0 | 112M | 1178G |
| VMamba-B | $512^2$ | 51.0 | 51.6 | 122M | 1170G |
| Spatial-Mamba-B | $512^2$ | **51.8** | **52.6** | 127M | 1176G |

## 5.2 OBJECT DETECTION AND INSTANCE SEGMENTATION ON COCO

**Settings.** We evaluate Spatial-Mamba in object detection and instance segmentation tasks using COCO 2017 dataset (Lin et al., 2014) and MMDetection library (Chen et al., 2019). We adopt Mask (He et al., 2017) and Cascade Mask R-CNN (Cai & Vasconcelos, 2018) as detector heads, apply Spatial-Mamba-T/S/B pre-trained on ImageNet-1K as backbones. Following common practices (Liu et al., 2021; 2024), we fine-tune the pre-trained models for 12 epochs ($1\times$ schedule) and 36 epochs with multi-scale inputs ($3\times$ schedule). During training, AdamW optimizer is adopted with an initial learning rate of 0.0001 and a batch size of 16.

**Results.** The results on COCO with Mask R-CNN are reported in Tab. 2, and the results with Cascade Mask R-CNN are provided in Appendix D. It can be seen that all variants of Spatial-Mamba outperform their competitors under different schedules. For $1\times$ schedule, Spatial-Mamba-T achieves a box mAP of 47.6 and a mask mAP of 42.9, surpassing Swin-T/VMamba-T by 4.9/0.3 in box mAP and 3.6/0.2 in mask mAP with fewer parameters and FLOPs, respectively. Similarly, Spatial-Mamba-S/B demonstrate superior performance to other methods under the same configuration. Furthermore, these trends of improved performance hold with the $3\times$ multi-scale training schedule. Notably, Spatial-Mamba-S achieves the highest box mAP of 50.5 and mask mAP of 44.6, surpassing VMamba-S with a considerable gain of 0.6 and 0.4, respectively.

## 5.3 SEMANTIC SEGMENTATION ON ADE20K

**Settings.** To assess the performance of Spatial-Mamba on semantic segmentation task, we train our models with the widely used UPerNet segmentor (Xiao et al., 2018) and MMSegmenation toolkit (Contributors, 2020). Consistent with previous work (Liu et al., 2021; 2024), we pre-train our model on ImageNet-1K, and use it as the backbone to train UPerNet on ADE20K dataset (Zhou et al., 2019). This training process encompasses 160K iterations with a batch size of 16. The AdamW is used as the optimizer with a weight decay of 0.01. The learning rate is set to $6 \times 10^{-5}$ with a linear learning rate decay. All the input images are cropped into $512 \times 512$.

Table 4: Ablation studies on Spatial-Mamba-T for neighbor set $\Omega$ and other module designs.

| Module design | #Param. (M) | FLOPs (G) | Throughput ↑ | Top-1 acc.↑ |
|---|---|---|---|---|
| Baseline | 25M | 4.4G | 1706 | 82.0 |
| $\Omega = \{3 \times 3\}$ | 25M | 4.4G | 1557 | 82.3 |
| $\Omega = \{5 \times 5\}$ | 25M | 4.5G | 1461 | 82.5 |
| $\Omega = \{\Omega_d \mid d = 1, 3, 5\}$ | 25M | 4.5G | 1209 | 82.7 |
| + Overlapped Stem | 27M | 4.5G | 1158 | 82.9 |
| + LPU | 27M | 4.5G | 1065 | 83.3 |
| + Re-Param | 27M | 4.5G | 1438 | 83.3 |
| + MESA | 27M | 4.5G | 1438 | 83.5 |

**Results.** The results on semantic segmentation are summarized in Tab. 3. Spatial-Mamba variants consistently achieve impressive performance. For instance, Spatial-Mamba-T attains a single-scale mIoU of 48.6 and a multi-scale mIoU of 49.4. This signifies an improvement of 1.5 mIoU over NAT-T and 0.6 mIoU over VMamba-T in single-scale input. This advantage is maintained with multi-scale input, where Spatial-Mamba-T is 1.0 mIoU higher than NAT-T and 0.6 mIoU higher than VMamba-T. Furthermore, Spatial-Mamba-B achieves the best performance with a multi-scale mIoU of 52.6.

## 5.4 ABLATION STUDIES

In this section, we ablate various key components of Spatial-Mamba-T on ImageNet-1K classification task in Tab. 4. Based on the configurations of Swin-T (Liu et al., 2021), we construct the baseline model as Spatial-Mamba-T but without the SASF module. This baseline uses a $4 \times 4$ convolution with a stride of 4 as stem layer and merges $2 \times 2$ neighboring patches for down-sampling.

**Neighbor set.** First, adjustments to the neighbor set $\Omega$ reveal that increasing the size from a $3 \times 3$ neighborhood to $5 \times 5$ results in a gradual improvement in accuracy, from 82.3% to 82.5%, albeit with a corresponding decrease in throughput. Furthermore, employing a broader dilated neighbor set with factors $d = 1, 3, 5$ increases the accuracy to 82.7%, while reducing throughput to 1158. This suggests a trade-off between speed and larger neighbor set.

**Local enhancement.** We replace the original non-overlapped stem and down-sampling layer with overlapped convolutions (refer to as 'Overlapped Stem' in Tab. 4), resulting in a gain of 0.2% in accuracy. We also incorporate the LPU (Guo et al., 2022), a depth-wise convolution placed at the top of each block and FFN, further increasing the accuracy by 0.4%. These modifications enrich the local information available between image patches before processing by the SASF module, enabling it to better capture structural dependencies.

**Optimization.** To further enhance the model efficiency, we implement the SASF module using re-parameterization techniques (Ding et al., 2022) and optimize the CUDA kernels. This accelerates the model by at least 30% and boosts the throughput from 1065 to 1438. Finally, integrating MESA (Du et al., 2022) to mitigate overfitting provides an additional 0.2% accuracy improvement.

In addition, we provide some qualitative results in Appendix E, an in-depth discussion about SASF in Appendix F, and a comparative analysis of the Effective Receptive Fields (ERF) (Ding et al., 2022) of various models is provided in Appendix G.

## 6 CONCLUSION

We presented in this paper Spatial-Mamba, a novel state space model designed for visual tasks. The key of Spatial-Mamba lied in the proposed structure-aware state fusion (SASF) module, which effectively captured image spatial dependencies and hence improved the contextual modeling capability. We performed extensive experiments on fundamental vision tasks of image classification, detection and segmentation. The results showed that with SASF, Spatial-Mamba surpassed the state-of-the-art state space models with only one signal scan, demonstrating its strong visual feature learning capability. We also analyzed in-depth the relationships of Spatial-Mamba with the original Mamba and linear attention, and unified them under the same matrix multiplication framework, offering a deeper understanding of the self-attention mechanism for visual representation learning.

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

# Appendix to "Spatial-Mamba: Effective Visual State Space Models via Structure-aware State Fusion"

In this appendix, we provide the following materials:

A  More visual results for SASF (referring to Sec. 4.1 in the main paper);

B  Architecture details of Spatial-Mamba and experimental settings (referring to Sec. 4.2 and Sec. 5.1 in the main paper);

C  Derivations of the SSM formulas of Mamba and Spatial-Mamba (referring to Sec. 4.3 in the main paper);

D  Results of Cascade Mask R-CNN detector head (referring to Sec. 5.2 in the main paper);

E  Visual results and comparisons for detection and segmentation tasks (referring to Sec. 5.2 and Sec. 5.3 in the main paper);

F  Further discussions about SASF (referring to Sec. 5.4 in the main paper);

G  Comparisons of effective receptive field (referring to Sec. 5.4 in the main paper).

## A  MORE VISUAL RESULTS

More visual comparisons between the original and fused structure-aware state variables are shown in Fig. 6. Due to the spatial modeling capability of SASF, the fused state variables demonstrate more accurate context information and superior structural perception across different scenarios.

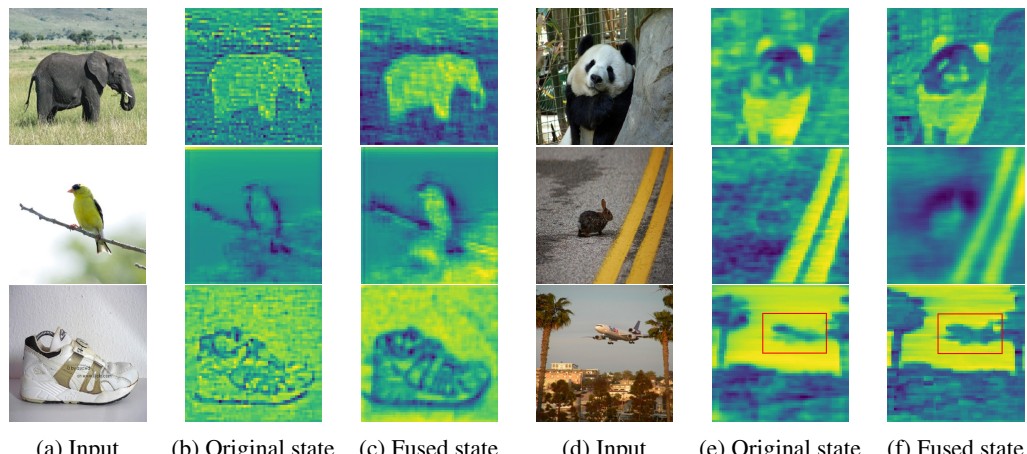

|         (a) Input |    (b) Original state |    (c) Fused state |         (d) Input |    (e) Original state |    (f) Fused state |

Figure 6: More visualizations of state variables before and after applying the SASF equation.

## B  MODEL ARCHITECTURES AND EXPERIMENT SETTINGS IN CLASSIFICATION

**Network architecture.** The detailed architectures of Spatial-Mamba models are outlined in Tab. 5. Following the common four-stage hierarchical framework (Liu et al., 2021; Han et al., 2024), we construct the Spatial-Mamba models by stacking our proposed Spatial-Mamba blocks at each stage. Specifically, an input image with resolution of $224 \times 224$ is firstly processed by a stem layer, which consists of Convolution (Conv), Batch Normalization (BN) and GELU activation function. The kernel size is $3 \times 3$ with a stride of 2 at the first and last convolution layers, and a stride of 1 for other layers. Each stage contains multiple Spatial-Mamba blocks, followed by a down-sampling layer except for the last block. The down-sampling layer consists of a $3 \times 3$ convolution with a stride of 2 and a Layer Normalization (LN) layer. Each block incorporates a structure-aware SSM layer and a Feed-Forward Network (FFN), both with residual connections. The structure-aware SSM contains a SASF branch with a 2D depth-wise convolution and a multiplicative gate branch with activation function, as illustrated in Sec. 4.2 of the main paper. The expand ratio of SSM is set to 2, doubling the

Table 5: Architectures of Spatial-Mamba models, where Linear refers to a linear layer, DWConv represents a depth-wise convolution layer, SASF module refers to Fig. 4 in Sec. 4.2 of the main paper, and FFN is a Feed-Forward Network.

| Layer | Output size | Spatial-Mamba-T | Spatial-Mamba-S | Spatial-Mamba-B |
|---|---|---|---|---|
| Stem | $56 \times 56$ | Conv $3 \times 3$ stride 2, BN, GELU; Conv $3 \times 3$ stride 1, BN; Conv $3 \times 3$ stride 2, BN | | |
| Stage1 | $28 \times 28$ | Spatial-Mamba Blocks $\begin{bmatrix} \text{Linear } 64 \to 128 \\ \text{DWConv } 128 \\ \text{SASF } 128 \\ \text{Linear } 128 \to 64 \\ \text{FFN } 64 \end{bmatrix} \times 2$ | Spatial-Mamba Blocks $\begin{bmatrix} \text{Linear } 64 \to 128 \\ \text{DWConv } 128 \\ \text{SASF } 128 \\ \text{Linear } 128 \to 64 \\ \text{FFN } 64 \end{bmatrix} \times 2$ | Spatial-Mamba Blocks $\begin{bmatrix} \text{Linear } 96 \to 192 \\ \text{DWConv } 192 \\ \text{SASF } 192 \\ \text{Linear } 192 \to 96 \\ \text{FFN } 96 \end{bmatrix} \times 2$ |
| | | Down Sampling Conv $3 \times 3$ stride 2, LN | | |
| Stage2 | $14 \times 14$ | Spatial-Mamba Blocks $\begin{bmatrix} \text{Linear } 128 \to 256 \\ \text{DWConv } 256 \\ \text{SASF } 256 \\ \text{Linear } 256 \to 128 \\ \text{FFN } 128 \end{bmatrix} \times 4$ | Spatial-Mamba Blocks $\begin{bmatrix} \text{Linear } 128 \to 256 \\ \text{DWConv } 256 \\ \text{SASF } 256 \\ \text{Linear } 256 \to 128 \\ \text{FFN } 128 \end{bmatrix} \times 4$ | Spatial-Mamba Blocks $\begin{bmatrix} \text{Linear } 192 \to 384 \\ \text{DWConv } 384 \\ \text{SASF } 384 \\ \text{Linear } 384 \to 192 \\ \text{FFN } 192 \end{bmatrix} \times 4$ |
| | | Down Sampling Conv $3 \times 3$ stride 2, LN | | |
| Stage3 | $7 \times 7$ | Spatial-Mamba Blocks $\begin{bmatrix} \text{Linear } 256 \to 512 \\ \text{DWConv } 512 \\ \text{SASF } 512 \\ \text{Linear } 512 \to 256 \\ \text{FFN } 256 \end{bmatrix} \times 8$ | Spatial-Mamba Blocks $\begin{bmatrix} \text{Linear } 256 \to 512 \\ \text{DWConv } 512 \\ \text{SASF } 512 \\ \text{Linear } 512 \to 256 \\ \text{FFN } 256 \end{bmatrix} \times 21$ | Spatial-Mamba Blocks $\begin{bmatrix} \text{Linear } 384 \to 768 \\ \text{DWConv } 768 \\ \text{SASF } 768 \\ \text{Linear } 768 \to 384 \\ \text{FFN } 384 \end{bmatrix} \times 21$ |
| | | Down Sampling Conv $3 \times 3$ stride 2, LN | | |
| Stage4 | $7 \times 7$ | Spatial-Mamba Blocks $\begin{bmatrix} \text{Linear } 512 \to 1024 \\ \text{DWConv } 1024 \\ \text{SASF } 1024 \\ \text{Linear } 1024 \to 512 \\ \text{FFN } 512 \end{bmatrix} \times 4$ | Spatial-Mamba Blocks $\begin{bmatrix} \text{Linear } 512 \to 1024 \\ \text{DWConv } 1024 \\ \text{SASF } 1024 \\ \text{Linear } 1024 \to 512 \\ \text{FFN } 512 \end{bmatrix} \times 5$ | Spatial-Mamba Blocks $\begin{bmatrix} \text{Linear } 768 \to 1536 \\ \text{DWConv } 1536 \\ \text{SASF } 1536 \\ \text{Linear } 1536 \to 768 \\ \text{FFN } 768 \end{bmatrix} \times 5$ |
| Head | $1 \times 1$ | Average pool, Linear 1000, Softmax | | |

number of channels. The SSM state dimension is set to 1 for better performance and efficiency. We modify the embedding dimension and number of blocks to build our Spatial-Mamba-T/S/B models.

**Settings for ImageNet-1K classification.** The Spatial-Mamba-T/S/B models are trained from scratch for 300 epochs using AdamW optimizer with betas set to (0.9, 0.999), momentum set to 0.9, and batch size set to 1024. The initial learning rate is set to 0.001 with a weight decay of 0.05. A cosine annealing learning rate schedule is adopted with a warm-up of 20 epochs. We adopt the common data augmentation strategies as in previous works (Liu et al., 2021; 2024). Moreover, label smoothing (0.1), exponential moving average (EMA) and MESA (Du et al., 2022) are also applied. The drop path rate is set to 0.2 for Spatial-Mamba-T, 0.3 for Spatial-Mamba-S and 0.5 for Spatial-Mamba-B.

**Implementation details.** The Spatial-Mamba models employ a hardware-aware selective scan algorithm adapted from the original Mamba framework, with modifications to the CUDA kernels for decoupling state transition and observation equations. The SASF module of Spatial-Mamba is implemented by the general matrix multiplication (GEMM) with optimized CUDA kernels.

## C   DERIVATION OF SSM FORMULAS

Based on the Mamba formulation provided in Sec. 3 of the main paper, the state transition equation in the recursive form can be rewritten as follows:

$$
\begin{aligned}
x_t &= \overline{\boldsymbol{A}}_t x_{t-1} + \overline{\boldsymbol{B}}_t u_t \\
&= \overline{\boldsymbol{A}}_t \left( \overline{\boldsymbol{A}}_{t-1} x_{t-2} + \overline{\boldsymbol{B}}_{t-1} u_{t-1} \right) + \overline{\boldsymbol{B}}_t u_t \\
&= \overline{\boldsymbol{A}}_t \left( \overline{\boldsymbol{A}}_{t-1} \left( \overline{\boldsymbol{A}}_{t-2} x_{t-3} + \overline{\boldsymbol{B}}_{t-2} u_{t-2} \right) + \overline{\boldsymbol{B}}_{t-1} u_{t-1} \right) + \overline{\boldsymbol{B}}_t u_t \\
&= \overline{\boldsymbol{A}}_t \overline{\boldsymbol{A}}_{t-1} \overline{\boldsymbol{A}}_{t-2} x_{t-3} + \overline{\boldsymbol{A}}_t \overline{\boldsymbol{A}}_{t-1} \overline{\boldsymbol{B}}_{t-2} u_{t-2} + \overline{\boldsymbol{A}}_t \overline{\boldsymbol{B}}_{t-1} u_{t-1} + \overline{\boldsymbol{B}}_t u_t \\
&= \Pi_{i=1}^{t} \overline{\boldsymbol{A}}_i x_0 + \Pi_{i=2}^{t} \overline{\boldsymbol{A}}_i \overline{\boldsymbol{B}}_1 u_1 + \cdots + \Pi_{i=t-1}^{t} \overline{\boldsymbol{A}}_i \overline{\boldsymbol{B}}_{t-2} u_{t-2} + \Pi_{i=t}^{t} \overline{\boldsymbol{A}}_i \overline{\boldsymbol{B}}_{t-1} u_{t-1} + \overline{\boldsymbol{B}}_t u_t
\end{aligned} \tag{7}
$$

By defining the initial state as zero, $x_0 = 0$, Eq. (7) can be expressed as:

$$x_t = \sum_{s \leq t} \overline{\boldsymbol{A}}_{s:t}^{\times} \overline{\boldsymbol{B}}_s u_s, \quad \text{where } \overline{\boldsymbol{A}}_{s:t}^{\times} := \begin{cases} \Pi_{i=s+1}^{t} \overline{\boldsymbol{A}}_i, & s < t \\ 1, & s = t \end{cases}. \tag{8}$$

We omit the term $\boldsymbol{D}_t u_t$ for simplicity. According to the observation equation, we can derive the final output $y_t = \sum_{s \leq t} \boldsymbol{C}_t \overline{\boldsymbol{A}}_{s:t}^{\times} \overline{\boldsymbol{B}}_s u_s$. Suppose the input vector is $u = [u_1, \cdots, u_L]^T \in \mathbb{R}^L$ with length $L$, the corresponding output vector is $y \in \mathbb{R}^L$. Then the above calculation can be written in matrix multiplication form, *i.e.*, $y = \boldsymbol{M} u$, where $\boldsymbol{M}$ is a structured lower triangular matrix and $\boldsymbol{M}_{ij} = \boldsymbol{C}_i \overline{\boldsymbol{A}}_{j:i}^{\times} \overline{\boldsymbol{B}}_j$.

Similarly, we can represent Spatial-Mamba in the same matrix transformation form. Based on the definition of Spatial-Mamba in Sec. 4.1 and Eq. (8), the SASF equation can be rewritten as $h_t = \sum_{k \in \Omega} \sum_{s \leq \rho_k(t)} \alpha_k \overline{\boldsymbol{A}}_{s:\rho_k(t)}^{\times} \overline{\boldsymbol{B}}_s u_s$. By multiplying $\boldsymbol{C}_t$, we can derive the final output of Spatial-Mamba as $y_t = \sum_{k \in \Omega} \sum_{s \leq \rho_k(t)} \alpha_k \boldsymbol{C}_t \overline{\boldsymbol{A}}_{s:\rho_k(t)}^{\times} \overline{\boldsymbol{B}}_s u_s$. It can also be concisely represented as a matrix multiplication form $y = \boldsymbol{M} u$, where $\boldsymbol{M}$ is a structured adjacency matrix and $\boldsymbol{M}_{ij} = \sum_k \boldsymbol{C}_i \overline{\boldsymbol{A}}_{j:\rho_k(i)}^{\times} \overline{\boldsymbol{B}}_j$.

## D  RESULTS OF CASCADE MASK R-CNN DETECTOR HEAD

Detailed results of object detection and instance segmentation on the COCO dataset with Cascade Mask R-CNN framework are reported in Tab. 6. We can see that Spatial-Mamba-T achieves a box mAP of 52.1 and a mask mAP of 44.9, surpassing Swin-T/NAT-T by 1.7/0.7 in box mAP and 1.2/0.4 in mask mAP with fewer parameters and FLOPs, respectively. Similarly, Spatial-Mamba-S demonstrates superior performance under the same configuration.

Table 6: Results of object detection and instance segmentation on the COCO dataset using Cascade Mask R-CNN (Cai & Vasconcelos, 2018) under $3\times$ schedule. FLOPs are calculated with input resolution of $1280 \times 800$.

| Backbone | $AP^b \uparrow$ | $AP_{50}^b \uparrow$ | $AP_{75}^b \uparrow$ | $AP^m \uparrow$ | $AP_{50}^m \uparrow$ | $AP_{75}^m \uparrow$ | #Param. | FLOPs |
|---|---|---|---|---|---|---|---|---|
| Swin-T | 50.4 | 69.2 | 54.7 | 43.7 | 66.6 | 47.3 | 86M | 745G |
| ConvNeXt-T | 50.4 | 69.1 | 54.8 | 43.7 | 66.5 | 47.3 | 86M | 741G |
| NAT-T | 51.4 | 70.0 | 55.9 | 44.5 | 67.6 | 47.9 | 85M | 737G |
| Spatial-Mamba-T | **52.1** | **71.0** | **56.5** | **44.9** | **68.3** | **48.7** | 84M | 740G |
| Swin-S | 51.9 | 70.7 | 56.3 | 45.0 | 68.2 | 48.8 | 107M | 838G |
| ConvNeXt-S | 51.9 | 70.8 | 56.5 | 45.0 | 68.4 | 49.1 | 108M | 827G |
| NAT-S | 52.0 | 70.4 | 56.3 | 44.9 | 68.1 | 48.6 | 108M | 809G |
| Spatial-Mamba-S | **53.3** | **71.9** | **57.9** | **45.8** | **69.4** | **49.7** | 101M | 794G |

## E  QUALITATIVE RESULTS

In this section, we present the visualization results of object detection and instance segmentation in Fig. 7, and present the results of semantic segmentation in Fig. 8. Compared with VMamba, our Spatial-Mamba demonstrates superior performance in both tasks, producing more accurate detection boxes segmentation masks, particularly in areas where local structural information is crucial. For example, in the second row of Fig. 7, VMamba mistakenly identifies the shoes on a skateboard as a person, probably because it observes the shoes from four directions independently and resembles them as a human. Our method avoids this mistake by simultaneously perceiving the shoes and their surrounding context. Similarly, in the semantic segmentation task, as shown in the second and third rows of Fig. 8, our approach achieves more precise structures of trees and doors. These results highlight the effectiveness of our proposed Spatial-Mamba in leveraging local structural information for better visual understanding.

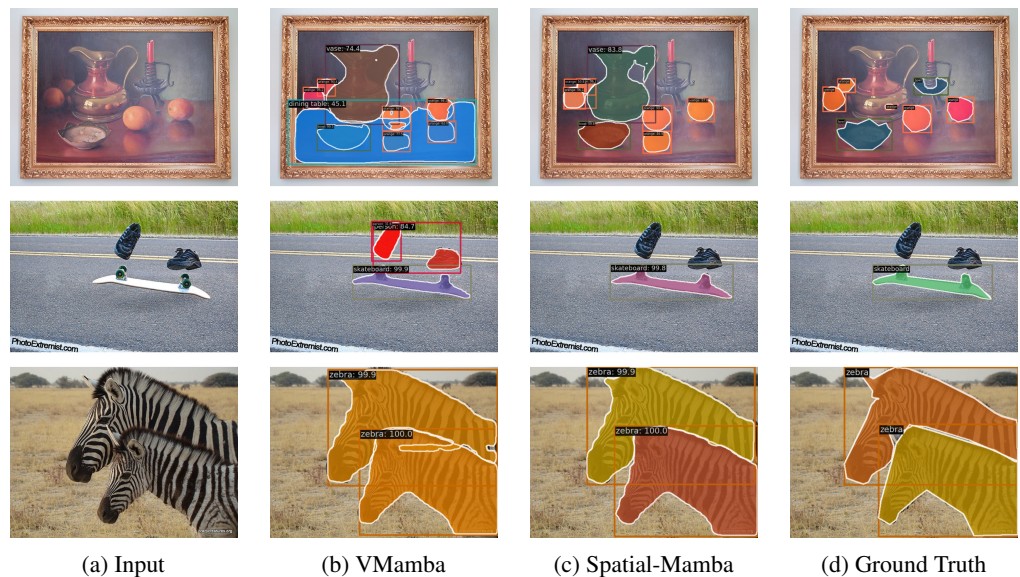

|  |  |  |  |
| --- | --- | --- | --- |
| (a) Input | (b) VMamba | (c) Spatial-Mamba | (d) Ground Truth |

Figure 7: Visualization examples of object detection and instance segmentation on COCO dataset with Mask R-CNN 1× schedule (He et al., 2017) by VMamba and Spatial-Mamba.

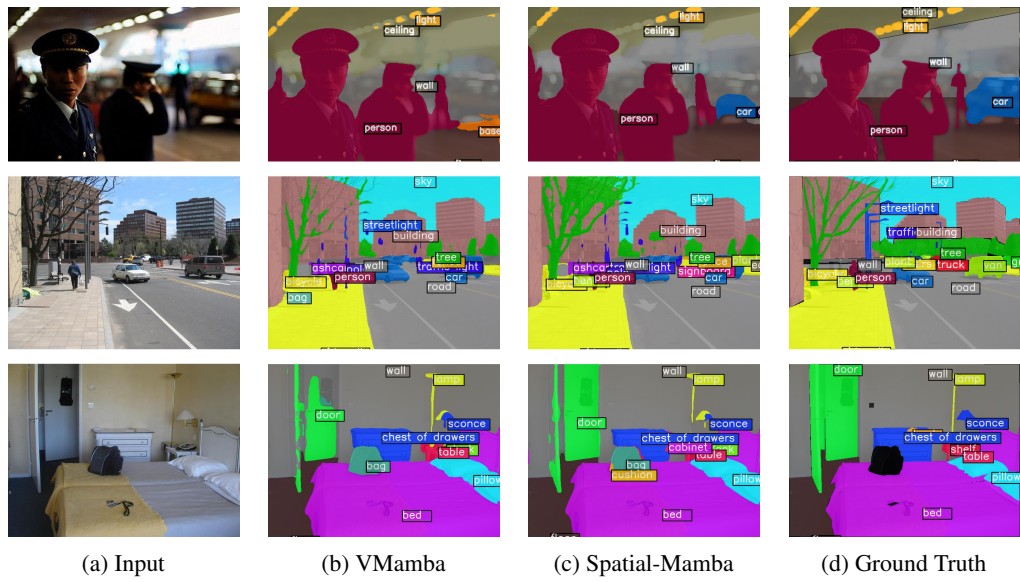

|  |  |  |  |
| --- | --- | --- | --- |
| (a) Input | (b) VMamba | (c) Spatial-Mamba | (d) Ground Truth |

Figure 8: Visualization examples of semantic segmentation on ADE20K dataset with single-scale inputs by VMamba and Spatial-Mamba.

## F  FURTHER DISCUSSIONS ABOUT SASF

**SASF extension.** To further explore and validate the spatial modeling capability of our proposed SASF, we adapt SASF into Vim and VMamba backbones, resulting in Vim+SASF (by replacing the middle class token with average pooling and setting state dimension to 1) and VMamba+SASF (directly integrated with SASF). Under the same training settings, the classification results on ImageNet-1K are presented in Tab. 7. It can be seen that by integrating with SASF, the bi-directional Vim model is improved by 0.5% in top-1 accuracy and the cross-scan VMamba is improved by 0.3%. These findings verify that SASF can even enhance the performance of multi-directional models, which have already incorporated (yet in an inefficient manner) spatial modeling operations.

Table 7: Comparison of classification performance on ImageNet-1K by integrating with SASF, where 'Throughput' is measured using an A100 GPU with an input resolution of $224 \times 224$.

| Method | Im. size | #Param. (M) | FLOPs (G) | Throughput↑ | Top-1 acc.↑ |
|---|---|---|---|---|---|
| Vim-Ti | $224^2$ | 6M | 1.4G | - | 71.7 |
| Vim-Ti+SASF | $224^2$ | 7M | 1.6G | - | **72.2** |
| VMamba-T | $224^2$ | 30M | 4.9G | 1686 | 82.6 |
| VMamba-T+SASF | $224^2$ | 31M | 5.1G | 1126 | **82.9** |

**Fusion operators.** There are also different choices of the operators that can be used for fusing state variables in SASF. We opt for depth-wise convolution due to its simplicity and computational efficiency. However, operators like dynamic convolution (Chen et al., 2020), deformable convolution (Dai et al., 2017), and attention mechanisms have also demonstrated superior performance in computer vision tasks. Therefore, we anticipate that they can be used as alternatives to the depth-wise convolution in our Spatial-Mamba. We simply train a Spatial-Mamba-T network with dynamic convolution and the results are shown Tab. 8. We can see that dynamic convolution yields a 0.2% performance gain in accuracy but significantly reduces throughput from 1438 to 907. This result suggests the potential advantages of more flexible fusion operators for SASF, but also highlights the importance of considering computational cost.

Table 8: Comparison of classification performance by Spatial-Mamba-T with depth-wise conv. and dynamic conv. on ImageNet-1K, where 'Throughput' is measured using an A100 GPU with an input resolution of $224 \times 224$.

| Method | Im. size | #Param. (M) | FLOPs (G) | Throughput↑ | Top-1 acc.↑ |
|---|---|---|---|---|---|
| Depth-wise Conv. | $224^2$ | 27M | 4.5G | 1438 | 83.5 |
| Dynamic Conv. | $224^2$ | 33M | 4.8G | 907 | 83.7 |

## G EFFECTIVE RECEPTIVE FIELD (ERF)

We compare the Effective Receptive Field (ERF) (Ding et al., 2022) of the center pixel on popular backbone networks before and after training, as shown in Fig. 9. The ERF values represent the contributions of every pixel on input space to the central pixel in the final output feature maps. To visualization, we randomly select 50 images from the ImageNet-1K validation set, resize them to a resolution of 1024×1024, and then calculate the ERF values with the auto-grad mechanism. Before training, our Spatial-Mamba-T initially exhibits a larger receptive field than other methods except DeiT-S due to neighborhood connectivity in the state space. After training, our method, along with DeiT-S, Vim-S, and VMamba-T, all demonstrate a global ERF. In addition, both Vim-S and VMamba-T exhibit noticeable accumulation contributions along either horizontal or vertical directions, which can be attributed to their multi-directional fusion mechanisms. In contrast, our unidirectional Spatial-Mamba-T effectively eliminates this directional bias.

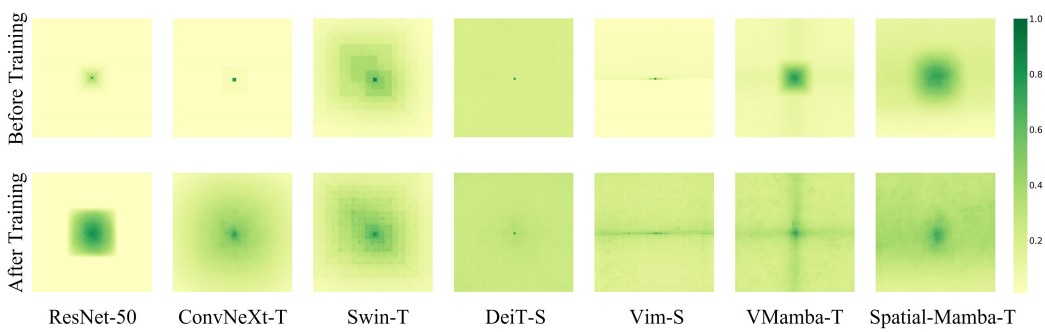

Figure 9: Comparison of Effective Receptive Field (ERF) among popular backbone networks.

