# OpenReview forum: "Spatial-Mamba: Effective Visual State Space Models via Structure-Aware State Fusion"
_ICLR.cc/2025/Conference — ICLR 2025 Poster_

### Official Review · Reviewer_P6PW · 2024-10-26

**Soundness:** 3
**Presentation:** 2
**Contribution:** 3
**Rating:** 6
**Confidence:** 4

**Summary:**

This paper introduces Spatial-Mamba, a novel state space model (SSM) designed for visual tasks, capable of effectively capturing spatial dependencies in images to enhance contextual modeling. Experimental results demonstrate that Spatial-Mamba outperforms existing SSMs and other mainstream models in areas such as image classification and semantic segmentation.

**Strengths:**

1. Improved upon existing SSMs by directly integrating spatial neighborhood dependencies into the state transition process.

2. Analyzed the connections between Spatial-Mamba, Mamba, and linear attention, unifying them under a matrix multiplication framework.

**Weaknesses:**

1. The paper does not mention the number of hidden states used in the SSM, and the ablation study lacks an analysis of the impact of different numbers of hidden states.

2. While the authors analyze the relationship between Spatial-Mamba, Mamba, and linear attention, they do not clarify the framework used to implement Spatial-Mamba, such as Mamba's hardware scanning algorithm or Mamba2's SSD framework.

3. In Mamba’s proposed SSM, hidden states do not inherently have spatial relationships. I suspect the authors intend to model the hidden states as 2D local observation points?

**Questions:**

1. Could you further explain how state variables are modeled in the state space to design the structure-aware state fusion?
2. Please provide details on the number of hidden states in each SSM layer and include relevant ablation experiments.
3. Is Spatial-Mamba implemented using the hardware scanning algorithm proposed by Mamba?

---

> ### Author Response · Authors · 2024-11-21
>
> We sincerely thank you for your constructive comments! Please find our responses to your concerns below.
>
> ---
> **Q1. Analysis on the number of hidden states.**
>
> Thank you for your insightful comments. We mainly follow the configurations of VMamba for fair comparison with the competing methods. The number of hidden states (the SSM state dimension, _i.e._, d\_state) is set to 1. The impact of the number of d\_state has been discussed in VMamba. It is found that gradually increasing d\_state from 1 to 4 slightly improves the performance, but the throughput drops significantly, indicating a negative impact on computational efficiency. We also analyzed the setting of d\_state in our early attempts (single directional scanning without SASF and MLP), and the results are shown in the Table below. We see that increasing d\_state from 16 to 32 improves the performance by 0.3\% in top-1 accuracy  on ImageNet-1K classification, but the throughput drops by 30\%. Therefore, in all our experiments, we set d\_state to 1 as the default value.
>
> Table4. Ablation study of the number of hidden states d\_state.
> |d\_state|Im. size|#Param. (M)|FLOPs (G)|Throughput↑|Top-1 acc.↑|
> | :--- | :---: | :---: | :---: | :---: | :---: |
> |16|224$^2$|19M|4.2G|1406|80.9|
> |32|224$^2$|20M|4.4G|1039|81.2|
>
> Mamba2 solves the above problem, which allows the use of a larger d\_state (64 or 128) while maintaining efficient throughput. Nonetheless, this is not our focus. In future work, we plan to extend our study by incorporating more ablation experiments on the number of hidden states under the Mamba2 architecture.
>
> ---
> **Q2. Clarification on the framework of SSM and implementation details.**
>
> Thank you for your valuable comments. We have provided more details about the framework used in Spatial-Mamba in the revised version. We employ a hardware-aware selective scan algorithm, which is based on the original Mamba framework with some modifications. Specifically, we made modifications in the CUDA kernels to decouple the state transition and the observation equations. Besides, the SASF module of Spatial-Mamba is implemented by the general matrix multiplication (GEMM) with optimized CUDA kernels.
>
> ---
>
> **Q3. The spatial relationships of hidden states.**
>
> As mentioned by this reviewer, the hidden states in the original Mamba, which was designed for 1D sequential data, cannot well encode the spatial relationships that are crucial for understanding 2D images. In 1D sequences, the relationships between elements are primarily defined by their sequential order. However, in 2D images, the relationships between pixels or states are much more complex, involving both horizontal and vertical dependencies. To address this limitation and enhance the expressive power of the state space, we establish connections between neighboring states in the 2D state space. This explicit modeling of local spatial relationships allows the model to capture the inherent structure of the image data, leading to a more informative and powerful representation. By considering the interactions between nearby states, our proposed method effectively integrates the spatial context that is neglected in the original Mamba.
>
> ---
>
> **Q4. Explanation on how state variables are modeled in SASF.**
>
> The key challenge in extending Mamba from 1D data to 2D images is how to establish spatial relationships between state variables. Intuitively, we could attend each state variable to its neighboring states, enabling spatial structure awareness in the state space. To achieve this, we introduce structure-aware state fusion equation between the state transition and observation equations (see Fig. 1(d) and Eq. (2) in the main paper). Unlike the original Mamba, where state variables are only connected to the previous state, SASF allows each state to capture information from neighboring states through dilated convolution, converting it to a structure-aware state variable. This SASF stage captures the spatial dependencies between neighboring states in 2D data. We will add more explanations in the revised manuscript.

---

> > ### Author Response · Authors · 2024-11-25
> > **Your feedback and further comments are appreciated.**
> >
> > Dear Reviewer P6PW,
> >
> > We sincerely thank you for your time in reviewing our paper and your constructive comments. We have posted our point-to-point responses in the review system. Since the public discussion phase will end very soon, we appreciate if you could read our responses and let us know your feedback and further comments.
> >
> > Best regards,
> > Authors of paper #2673

---

> > > ### Comment · Reviewer_P6PW · 2024-11-25
> > >
> > > Thanks for the response. Most of my concerns have been addressed and I recommend accepting this work.

---

> > > > ### Author Response · Authors · 2024-11-25
> > > >
> > > > We sincerely thank this reviewer for the positive feedback and for accepting our work!

---

### Official Review · Reviewer_CLPg · 2024-10-29

**Soundness:** 4
**Presentation:** 4
**Contribution:** 2
**Rating:** 6
**Confidence:** 2

**Summary:**

This manuscript proposes to aggregate neighborhood features in state-space models for more efficient state-space modeling. The experiments are convincing, and the manuscript is well-organized.

**Strengths:**

+ The proposed fusing operation seems to have a decent performance gain onto the baseline.
+ The manuscript is well-organized.
+ The ablation studies are clear and convincing.

**Weaknesses:**

+  Novelty. The proposed fusion seems to be a re-invention and re-introduce of convolution on the feature map of state-space modeling but is claimed as "Structure-Aware State Fusion". The LPU, overlapped stem,  re-parameterization  and MESA are all proposed in previous works. The technical contribution seems to be limited.
+ Clearness. The fusion seems to be applied after scanning. However, in Eq(2), the $x_t$ is the same in scanning and fusion. The notation here is not clear. It would be better to use $x_t^{\prime}$ in the fusion formula if my understanding is correct.

**Questions:**

Please refer to weakness part where I have expressed my concerns.

---

> ### Author Response · Authors · 2024-11-21
>
> We sincerely thank you for your constructive comments and recognition on our work! We hope our following responses can address your concerns.
>
> ---
> **Q1. Clarification on novelty.**
>
> Thanks for your comments. We are sorry if we did not explain our contributions clearly enough in the manuscript, and we would like to clarify the novelties of our work in this response.
>
> The SASF module is not a re-introduction of convolution, but an innovative design on how we can effectively fuse states to capture complex spatial relationships while preserving the advantages of the original Mamba. As indicated by this reviewer, LPU, overlapped stem, re-parameterization and MESA have been explored in prior works, and we just adopt them in the implementation of our model to further improve its performance. Our main contribution lies in the visual state space modeling through SASF to capture spatial dependencies in 2D data. As we responded to the third question of reviewer Q4w4, SASF can be implemented by different operators beyond depth-wise convolution. In addition, we have analyzed the impact of overlapped stem, LPU, re-parameterization, and MESA in the ablation study. The results showed that that even without using these modules, Spatial-Mamba-T still outperforms the baseline by 0.7\% and VMamba-T (with overlapped stem) by 0.3\% in top-1 accuracy on ImageNet-1K. This validates the effectiveness of SASF.
>
> ---
> **Q2. Clarification on Eq. (2).**
>
> Thanks for your comments. The SASF is applied after the scanning phase but before the observation equation in the model. We would like to clarify that the notation used in Eq. (2) is accurate and appropriate. For your convenience, we briefly summarize the processes of Mamba and Spatial-Mamba in this response. The standard Mamba workflow processes input $u_t$ through the state transition phase (the scanning process) to generate $x_t$, which is then passed directly into the observation phase to produce the output $y_t$ $\mathbf{(}u_t \to x_t \to y_t \mathbf{)}$. In contrast, our Spatial-Mamba workflow introduces an additional step: input $u_t$ is processed through the scanning phase to generate $x_t$, which then goes through the SASF phase to produce the fused state $h_t$. Finally, $h_t$ is passed into the observation equation to yield $y_t$ $\mathbf{(}u_t \to x_t \to {\color{red} h_t} \to y_t \mathbf{)}$. Therefore, the notation used in Eq. (2) is accurate and appropriate.

---

> > ### Comment · Reviewer_CLPg · 2024-11-21
> > **Reponse**
> >
> > Thanks for your response! After reading the response and the review from other reviewers, I'd like to raise my score to 6. However, I still think the notation in Eq.(2) is inaccurate. Isn't the $x_t$s changes after the scanning process?

---

> > > ### Author Response · Authors · 2024-11-21
> > >
> > > We sincerely thank this reviewer for the positive feedback, and we are sorry for not explaining Eq. (2) clearly enough. To further clarify, let’s consider the following toy example. Assuming there are three input image patches {$u_1, u_2, u_3$} and an initial zero state variable $x_{0}$, with the state transition equation in Eq. (2), we can recursively calculate the state variables as $x_1=Ax_0+Bu_1=Bu_1$, $x_2=Ax_1+Bu_2=ABu_1+Bu_2$ and $x_3=Ax_2+Bu_3=A(ABu_1+Bu_2)+Bu_3$ (here the parameters $A$ and $B$ are simplified). Subsequently, these state variables {$x_1, x_2, x_3$} are passed to our proposed SASF to generate the fused state $h_t$. For simplicity, we set the fusion rule as $h_1 = x_1 + x_2$, $h_2 = x_2 + x_3$, and $h_3 = x_3 + x_1$, obtaining the fused states {$h_1, h_2, h_3$}. It is important to note that this fusion process is performed after the entire scanning process, ensuring that the state variables {$x_1, x_2, x_3$} remain unchanged, as we described in Eq. (2). We hope that our explanation can address this reviewer’s question.

---

> > > > ### Comment · Reviewer_CLPg · 2024-11-21
> > > > **Response**
> > > >
> > > > Thanks for the response. I'm now persuaded by the authors. My concerns are all dissolved and I'm opt-in for accepting this work.

---

> > > > > ### Author Response · Authors · 2024-11-22
> > > > >
> > > > > We sincerely thank this reviewer for agreeing with us and accepting our work!

---

### Official Review · Reviewer_Q4w4 · 2024-11-01

**Soundness:** 3
**Presentation:** 4
**Contribution:** 3
**Rating:** 8
**Confidence:** 4

**Summary:**

This paper introduces Spatial-Mamba, a  state space models visual backbone designed for 2D visual tasks. It mainly introduce a multi-scale depthwise conv layer after the mamba operators to fuse the contextual information from different spatial locations. The author conducts theoretical analysis. Extensive experiements like image classification, dense prediction tasks are conducted.  The results show it outperforms the baselines and achieve competitive resutls.

**Strengths:**

1. The motivation is reasonbal and good. The vision mamba models suffers from processing  2D visual data due to the complexity of spatial structures. It heavily relys  on the 1D scanning strategy to transform 2D format into 1D sequence. The author apply conv-based layers to fuse the contextual information  in spatial domain, which is complemenary to the Mamba based operator.

2. The experimetanl results are good. The accuracy in imageNet shows the improvement. The speed is still competitive as the conv operators are optimized, thus bringing less latency.

3. The overall method is simple and easy to understand. Applying conv layers as a compelmentary module to improve the spatial modeling capability of the mamba is simple and direct.

**Weaknesses:**

1. The method seems to be a combination of Mamba and a classical module (depthwise conv).  It is also mentioned in No.3 point in Strengths. I would recommned author to validate/explore the spatial modeling enhancement methods more throughly. For example, applying the multi-scale conv layer to the original bi-directional Mamba (Vim) and cross-scan mamba (VMamba) to see the improvements. THough the one-direction scan and conv layer can aggregate enough spatial contextual information.

2. The analysis of linear attention, mamba and the proposed method in section 4.3 is good. But the method proposed is rather straightforward. It seems the method doesnot improve the weakness of Mamba from the in-depth analysis. It first uses the Selective scan in Mamba to calculate the hidden states (one direction), following a depthwise conv to fuse the hidden states, then calculate the output y. It is not known the improvements over the method that adding more direcitons into the selective scan and computing the output.

3. The modules/operators to fuse the contextual information can have mutiple choices. The paper chooses depthwise conv. Conv operator may have inductive bias, which aggregate the spatial information by a fixed pattern. I would see more explorations, which could improve the paper.

**Questions:**

Please see weakness.

---

> ### Author Response · Authors · 2024-11-21
>
> We sincerely thank you for your constructive comments! Please find our response to your concerns below.
>
> ---
> **Q1. Further exploration of spatial modeling ability.**
>
> Thanks for your constructive comments. We'd like to clarify that our proposed Spatial Mamba is not a simple combination of Mamba and depth-wise conv, while depth-wise conv is an efficient way to implement our proposed SASF. Our main contribution lies in the innovative design of SASF, which considers both the global long-range dependencies and local spatial relationships within the state space. As mentioned in our manuscript, while existing multi-directional models (like those using bi-directional and cross-scan models) possess certain spatial modeling capability, they introduce much more additional computational costs, limiting their overall performance. Our Spatial-Mamba, with a single scan, achieves both improved accuracy and faster inference than existing visual Mamba models.
>
> Based on this reviewer's suggestion, to further explore and validate SASF's spatial modeling capability, we adapt SASF into Vim and VMamba backbones, resulting in Vim+SASF (by replacing the middle class token with average pooling and setting state dimension to 1) and VMamba+SASF (directly integrated with SASF). Under the same training settings, the classification results on ImageNet-1K are presented in the following Table:
>
> Table2. Comparison of classification performance on ImageNet-1K.
> |Method|Im. size|#Param. (M)|FLOPs (G)|Throughput↑|Top-1 acc.↑|
> | :--- | :---: | :---: | :---: | :---: | :---: |
> |Vim-Ti| 224$^2$| 6M|1.4G|-|71.7|
> |Vim-Ti+SASF|224$^2$|7M|1.6G|-|**72.2**|
> |VMamba-T|224$^2$|30M|4.9G|1686|82.6|
> |VMamba-T+SASF|224$^2$|31M|5.1G|1126|**82.9**|
>
> It can be seen that by integrating with our proposed SASF, the bi-directional Vim model is improved by 0.5\% in top-1 accuracy and the cross-scan VMamba is improved by 0.3\%. These findings verify that SASF can even enhance the performance of multi-directional models, which have already incorporated (yet in an inefficient manner) spatial modeling operations. We will supplement these experiments in the ablation study of the revised manuscript.
>
> ---
> **Q2. In-depth analysis of the proposed method.**
>
> Thanks for your comments. We'd like to clarify that our proposed Spatial-Mamba indeed overcomes the weakness of original Mamba, at least to certain extent, in processing 2D visual data by introducing the SASF equation, while we provide a simple yet and effective solution to implement SASF via depth-wise convolution. In **Sec. 1 and Fig. 1**, **Sec. 4.1 and Fig. 3**, and **Sec. 4.3 and Fig. 5** of the manuscript, we have analyzed the weaknesses of Mamba in adapting to 2D data and how Spatial-Mamba can address them. In specific:
> - **Sec. 1 and Fig. 1** outline the limitations of Mamba’s directional scanning in preserving 2D spatial relationships and its computational inefficiencies.
> - **Sec. 4.1 and Fig. 3** demonstrate how Spatial-Mamba explicitly models spatial relationships within the state space, effectively overcoming Mamba’s limitations in processing 2D data.
> - **Sec. 4.3 and Fig. 5** present a detailed comparative analysis, showing how Spatial-Mamba achieves superior spatial representation by leveraging weighted summation over a broader neighborhood, as visualized through adjacency matrices.
>
> To further clarify, we have updated Fig. 5 in the manuscript for better illustration. By rearranging the elements of certain rows of $\mathbf{M}$ (indicated by the red arrow) into image space, we illustrate the activation map of the corresponding patch (marked with a red star) on the right side of the figure. This visualization demonstrates that linear attention focuses on a limited area, while Mamba captures a broader region due to its long-range context modeling. Our Spatial-Mamba not only largely extends the range of context modeling but also enables spatial structure modeling, effectively identifying relevant regions even when they are distant from each other.

---

> > ### Author Response · Authors · 2024-11-21
> >
> > **Q3. More attempts on fusion operators.**
> >
> > Thanks for the suggestion. This reviewer is  correct. There are different choices of the operators that can be used for fusing state variables. We opt for depth-wise convolution due to its simplicity and computational efficiency. However, operators like dynamic convolution, deformable convolution, and attention mechanisms have also demonstrated superior performance in computer vision tasks. Therefore, we anticipate that they can be used as alternatives to the depth-wise convolution in our Spatial-Mamba. We just quickly trained a Spatial-Mamba-T network with dynamic convolution and the results are shown below:
> >
> > Table2. Comparison of classification performance by Spatial-Mamba-T with depth-wise conv. and dynamic conv. on ImageNet-1K.
> > |Method|Im. size|#Param. (M)|FLOPs (G)|Throughput↑|Top-1 acc.↑|
> > |:---|:---:|:---:|:---:|:---:|:---:|
> > |Depth-wise Conv.| 224$^2$| 27M|4.5G|1438|83.5|
> > |Dynamic Conv.| 224$^2$| 33M|4.8G|907|83.7|
> >
> > We can see that dynamic convolution yields a 0.2\% performance gain in accuracy but reduces throughput from 1438 to 907. This result suggests the potential advantages of more flexible fusion operators for SASF, but also highlights the importance of considering computational cost.

---

> > > ### Author Response · Authors · 2024-11-25
> > > **Your feedback and further comments are appreciated.**
> > >
> > > Dear Reviewer Q4w4,
> > >
> > > We sincerely thank you for your time in reviewing our paper and your constructive comments. We have posted our point-to-point responses in the review system. Since the public discussion phase will end very soon, we appreciate if you could read our responses and let us know your feedback and further comments.
> > >
> > > Best regards,
> > > Authors of paper #2673

---

> > > > ### Comment · Reviewer_Q4w4 · 2024-11-25
> > > >
> > > > Thanks for the rebuttal, most of my concerns are well-addressed and I raise my rating to 8

---

> > > > > ### Author Response · Authors · 2024-11-25
> > > > >
> > > > > We sincerely thank this reviewer for the positive feedback!

---

### Official Review · Reviewer_ZoEw · 2024-11-03

**Soundness:** 3
**Presentation:** 4
**Contribution:** 2
**Rating:** 8
**Confidence:** 3

**Summary:**

Spatial-Mamba introduces a "structure-aware state fusion" (SASF) mechanism using dilated convolutions in the Mamba architecture. This mechanism claims to add neighborhood connectivity directly in the state-space of SSM. The method is benchmarked on common image benchmarks, i.e. segmentation, classification, and detection using common datasets.

**Strengths:**

SSMs do have the problem of spatial sense. While the recent papers of ViM and VMamba try to solve them in their way, the solution feels rather underwhelming. The idea of introducing dilated convolutions as a means of capturing structural dependency in images is quite intriguing. I like the idea and the authors have done a good job of explaining most of the paper and related works. It is written clearly and concisely.

**Weaknesses:**

There are not many weaknesses in this paper. Apart from Figure 5 (and qualitative and erf figures in supp.), the paper could improve on convincing the reader with visuals if the resultant SSM with SASF mechanism extracts structural dependencies from the images.
Minor mistakes exist in the paper; specifically, on lines 199 and 200, equation 2 lacks a definition for alpha_k. It is only on line 212 that the alpha_k is mentioned as weights. Also, it is not clear from the text how exactly is this linear weighting being applied/calculated (on line 99, it mentions dilated convolutions to re-weight and merge states but it will be nice to add some clarity after equation 2 is introduced).

**Questions:**

The results for Cascade Mask R-CNN seem to be missing...
Minor: On line 204, it is mentioned "considering both temporal and spatial information", but what exactly is the temporal information here? Or is it a typo?

Overall, the paper is sound and presents a good idea for an existing problem. I am open to adjusting my evaluation should any significant concerns be raised by other reviewers that I may have overlooked.

---

> ### Author Response · Authors · 2024-11-21
>
> Thank you for your insightful comments! Please find our responses below.
>
> ---
> **Q1. More visual results.**
>
> We sincerely thank this reviewer for the recognition on our work. Actually, in Fig. 3 of our manuscript, we have provided a visualization on the differences in state variables before and after our SASF, highlighting SASF's effectiveness in spatial modeling. Based on the suggestion of this reviewer, we have provided more visual results in Fig. 1 of the updated supplementary materials. We can see that the fused state variables show more accurate context information and structural perception capabilities across different scenarios.
>
> ---
> **Q2. Definition of $\alpha_k$ and clarification on the linear weighting.**
>
> We are sorry for the missing description of $\alpha_k$ in Eq. (2). $\alpha_k$ represents the learnable weight used in our weighting method. As illustrated in Fig. 1(d) of the main paper, neighboring state variables (shown in blue and orange colors) are aggregated using these weights to produce a fused state variable (shown in red color). This fusion process is efficiently implemented using dilated convolutions. We will add these descriptions and make the definition of $\alpha_k$ and the calculation of linear weighting clearer in the revised manuscript.
>
> ---
> **Q3. Results of Cascade Mask R-CNN.**
>
> Thanks for the suggestion. We followed previous works (VMamba, LocalMamba) to use Mask R-CNN as the detector head in the experiments for a fair comparison with them. As suggested by this reviewer, we further performed experiments on Spatial-Mamba under the Cascade Mask R-CNN framework. We adopt the same configurations and hyper-parameter settings as Swin Transformer and fine-tune the pre-trained Spatial-Mamba-T/S models for 36 epochs (3x schedule) with Cascade Mask R-CNN detector head on the COCO dataset. The results are shown in the Table below:
>
> Table1. Cascade Mask R-CNN - 3x schedule
> | Backbone | AP$^b$↑ | AP$^b_{50}$↑ | AP$^b_{75}$↑ |AP$^m$↑ | AP$^m_{50}$↑ | AP$^m_{75}$↑ |#Param.| FLOPs|
> | :----------- | :--------------: | :--------------: | :--------------: | :--------------: | :--------------: | :--------------: | :--------------: | :--------------: |
> | Vim-Ti*         | 45.7 | 63.9 | 49.6 | 39.2 | 60.9 | 41.7 | - | -|
> | Swin-T         | 50.4 | 69.2 | 54.7 | 43.7 | 66.6 | 47.3 | 86M | 745G|
> | ConvNeXt-T    | 50.4 | 69.1 | 54.8 | 43.7 | 66.5 | 47.3 | 86M | 741G|
> | NAT-T         | 51.4 | 70.0 | 55.9 | 44.5 | 67.6 | 47.9 | 85M | 737G|
> | Spatial-Mamba-T | **52.1**  | **71.0**  | **56.5**  | **44.9**  | **68.3**  | **48.7**  |  84M |  740G|
> | Swin-S         | 51.9 | 70.7 | 56.3 | 45.0 | 68.2 | 48.8 | 107M | 838G|
> | ConvNeXt-S    | 51.9 | 70.8 | 56.5 | 45.0 | 68.4 | 49.1 | 108M | 827G|
> | NAT-S         | 52.0 | 70.4 | 56.3 | 44.9 | 68.1 | 48.6 | 108M | 809G|
> | Spatial-Mamba-S |  **53.3** | **71.9**  | **57.9**  | **45.8**  | **69.4**  | **49.7**  |  101M |  794G|
>
> We can see that Spatial-Mamba-T achieves a box mAP of 52.1 and a mask mAP of 44.9, surpassing Swin-T/NAT-T by 1.7/0.7 in box mAP and 1.2/0.4 in mask mAP with fewer parameters and FLOPs, respectively. Similarly, Spatial-Mamba-S demonstrates superior performance under the same configuration. We will add these results in the revision.
>
> _(Note$^*$: The results of Vim shown in the first row of the table are for reference only, because in the original paper of Vim, only the results of the Vim-tiny variant (7M) with Cascade Mask R-CNN are reported, and the training framework is also different from our 3x schedule.)_
>
> ---
> **Q4. Explanation on the temporal information.**
>
> We are sorry for the confusion caused. As mentioned in the manuscript, compared with the original Mamba, we enhance its spatial modeling capabilities by introducing a structure-aware state fusion (SASF) equation after the state transition equation. Our intention of using "temporal" here is to express that we retain the advantages of the original Mamba in processing temporal data, such as its ability to capture long-range and selective information. We will revise the sentence "by considering both temporal and spatial information" to "by considering both the global long-range and the local spatial information" in order to avoid possible misunderstandings.

---

> ### Comment · Reviewer_ZoEw · 2024-11-22
>
> I appreciate the author’s response.
>
> For Fig. 3, I am unsure how exactly in Fig. 3 (b) and Fig. 3 (c) the high mean of state variables in non-interesting regions can be interpreted as the method's effectiveness.
>
> Thank you for including the 3x schedule results.
>
> My other concerns, as well as those raised by other reviewers, have been mostly addressed. I will maintain my recommendation for accepting this paper.

---

> > ### Author Response · Authors · 2024-11-23
> >
> > We sincerely thank the reviewer for the positive feedback and recognition of our work. Regarding Fig. 3, we would like to clarify that the visualization represents the mean of the state variables, rather than traditional activation maps. These mean state variables reflect the spatial correlations of the input image features within the state space. Specifically, Fig. 3(b) shows that the original state variables fail to clearly distinguish the foreground from the background (e.g., the dragonfly wing region). In contrast, Fig. 3(c) demonstrates that, thanks to SASF’s ability to capture richer spatial structural information, the fused state variables can more effectively separate the foreground from the background (e.g., more clearly distinguishing the dragonfly wings from the background region).
> >
> > Regarding why the state variables in the background regions exhibit higher values in Fig. 3(b) and Fig. 3(c), we'd like to explain it from the following points. Since the state variables will go through the observation equation to produce the final activation responses, their relationships in the state space can be either positive (higher values in the foreground) or negative (higher values in the background). We have provided additional visualizations of different scenarios in Fig. 1 of the updated supplementary materials, from which we can see that regardless of the correlation type, the fused state variables consistently capture richer spatial structure and contextual information, enabling a clearer separation of the foreground and background. We hope this explanation can resolve this reviewer’s concerns.

---

> > > ### Comment · Reviewer_ZoEw · 2024-11-24
> > >
> > > Thank you very much for the detailed explanation. It surely alleviated my remaining concerns.

---

### Meta-Review · Area_Chair_tSrN · 2024-12-19

**Metareview:**

This paper introduces Spatial-Mamba, a backbone based on State Space Modeling (SSM) specifically designed for 2D vision tasks. It establishes neighborhood connectivity within the state space through a method known as structure-aware state fusion.

The strengths of the paper include a clear motivation, solid experimental results, and a straightforward method that is easy to understand. However, there are some concerns regarding the novelty of the work and the lack of comprehensive ablation studies.

The paper received generally positive reviews with scores of 8, 8, 6, and 6, resulting in an average score of 7. Since all reviewers acknowledged the paper's strengths, the Area Chair recommends its acceptance.

**Additional Comments On Reviewer Discussion:**

The reviewers generally appreciated the paper's strong motivation, intuitive solution, and promising empirical results. However, some concerns were raised regarding the novelty of the approach and the choice of fusion operation. The authors addressed these questions effectively. Additionally, reviewers pointed out some ambiguities and missing details in the paper’s explanations and experiments. The authors also responded well to these concerns, and the final manuscript can incorporate the suggested improvements. Furthermore, several constructive suggestions were made by the reviewers, and the authors conducted further analysis and experiments that enhanced the paper's quality. Overall, the advantages of this work outweigh its shortcomings, so the Area Chair recommends accepting the paper.

---

### Decision · Program_Chairs · 2025-01-22

Accept (Poster)